# Improved jellyfish gelatin quality through ultrasound-assisted salt removal and an extraction process

Artima Lueyot[1], Benjamaporn Wonganu[2], Vilai Rungsardthong[1,3], Savitri Vatanyoopaisarn[1,3], Pokkwan Hutangura[1], Pisit Wongsa-Ngasri[4], Sittiruk Roytrakul[5], Sawanya Charoenlappanit[5], Tao Wu[6], Benjawan Thumthanaruk[1,3]*

1 Department of Agro-Industrial, Food, and Environmental Technology, Faculty of Applied Science, King Mongkut's University of Technology North Bangkok, Bangkok, Thailand, 2 Food and Agro-Industry Research Center, King Mongkut's University of Technology North Bangkok, Bangkok, Thailand, 3 Department of Biotechnology, Faculty of Applied Science, King Mongkut's University of Technology North Bangkok, Bangkok, Thailand, 4 Fishery Technological Development Division, Department of Fisheries, Ministry of Agriculture and Cooperatives, Bangkok, Thailand, 5 Functional Ingredients and Food Innovation Research Group, National Center for Genetic Engineering and Biotechnology, National Science and Technology Development Agency, Pathumthani, Thailand, 6 Department of Food Science, The University of Tennessee, Knoxville, Tennessee, United States of America

* benjawan.t@sci.kmutnb.ac.th

**Data Availability Statement:** All relevant data are within the manuscript and its Supporting information files.

## Abstract

The use of by-products of salted jellyfish for gelatin production offers valuable gelatin products rather than animal feed. Several washes or washing machines have reported removing salt in salted jellyfish. However, the green ultrasound technique has never been reported for the desalination of salted jellyfish. The objectives were to determine how effectively the raw material's salt removal was done by combining the traditional wash and then subjected to the ultrasonic waves in a sonication bath for 20–100 min. For gelatin production, the ultrasonicated jellyfish by-products were pretreated with sodium hydroxide and hydrochloric acid, washed, and extracted with hot water for 4, 6, and 8 h. Results showed that the increased duration of ultrasound time increased the desalination rate. The highest desalination rate of 100% was achieved using 100 min ultrasonic time operated at a fixed frequency (40 kHz) and power (220 W). The jellyfish gelatin extracted for 4, 6, and 8 h showed gel strengths in 121–447, 120–278, and 91–248 g. The 80 min ultrasonicated sample and hot water extraction for 8 h (JFG80-8) showed the highest gel yield of 32.69%, with a gel strength of 114.92 g. Still, the 40 min ultrasonicated sample with 4 h of extraction delivered the highest gel strength of 447.01 g (JFG40-4) and the lower yield of 10.60%. The melting and gelling temperatures of jellyfish gelatin from ultrasonicated samples ranged from 15–25°C and 5–12°C, which are lower than bovine gelatin (BG) and fish gelatin (FG). Monitored by FITR, the synergistic effect of extended sonication time (from 20–100 min) with 4 h extraction time at 80 °C caused amide I, II, and III changes. Based on the proteomic results, the peptide similarity of JFG40-4, having the highest gel strength, was 17, 23, or 20 peptides compared to either BG, FG, or JFG100-8 having the lowest gel strength. The 14 peptides were similarly found in all JFG40-4, BG, and FG samples. In conclusion, for the first time in this report, the

**Funding:** Artima received funding from the Department of Agro-Industrial, Food and Environmental Technology and College of graduate, King Mongkut's University of Technology North Bangkok. This work was financially supported by Thailand Science Research and Innovation (TSRI) (No. PHD59I0041). There was no additional external funding received for this study. The funders had no role in study design, data collection and analysis, decision to publish, or preparation of the manuscript.

**Competing interests:** The authors have decleared that no competing interests exist.

improved jellyfish gel can be achieved when combined with traditional wash and 40 min ultrasonication of desalted jellyfish and extraction time of 4 h at 80 ˚C.

## Introduction

Marine gelatin, one of the biomaterials involved in food and medicinal research, is denatured collagen produced from acid, alkaline, or enzyme hydrolysis. Gelatin is a crucial ingredient for the food, pharmaceutical, medical, biomedical, and photography industries, focusing on versatile applications. Based on market research, the global market size of gelatin from porcine, bovine skin, bovine bone, fish, and poultry sources will reach 3.6 billion US$ by 2023, at an average rate increasing rate of 6.6%, calculated from the data of 2018 [1]. Although bovine and porcine gelatins have been marketable and have gained a higher market share than marine gelatin, to date, the safety awareness of diseases including bovine spongiform encephalopathy (BSE) and foot-and-mouth disease, as well as strict religious rules, have caused a significant impact resulting in increased use of marine gelatin in all applications [2]. The advantages of marine gelatin, especially jellyfish gelatin, include halal and kosher certification, the absence of genetically modified organisms, and no known allergies [3]. Commercial fish gelatin is mainly produced from by-products of tilapia skin or scale. In contrast, other sources of by-products such as fins, frames, heads, shells, skin, and viscera from the fishery industry can also be used [4]. Research has been carried out on gelatin from several marine by-products as raw materials, including salmon skin [5], jellyfish [6, 7], fish [8], and solid waste from surimi processing [9]. Regardless of by-product usage, edible jellyfish by-products derived from the trimming of salted jellyfish for export have recently gotten attention for gelatin research [6, 7, 10, 11].

For over a century, edible jellyfish, zooplankton rich in collagen, have been welcome in Asian cuisine [12, 13]. The collagen content of the edible species of jellyfish such as *Nemopilema nomurai*, *Stomolophus meleagris*, *Catostylus tagi*, *Rhizostoma pulmo*, *Rhopilema esculentum*, *Rhopilema hispidum*, *Lobonema smithii*, and *Acromitus flagellatus* varies depending on species and portion of analysis [14–18]. In Thailand, the edible jellyfish *Lobonema smithii* has been exported chiefly to Japan and South Korea, with an average export value of close to 15 million US$ (1 US$ = 32 Baht) from 2010–2021 [19]. The global jellyfish business is expected to be 100 million US$, with trading with at least 23 countries around the globe [20]. In Asia, the processing of salted jellyfish has been reported by using a salt mixture containing salt, alum, and soda with slight differences in jellyfish species, duration of salting, and quantity of salt mixture used in the process [12, 21–28]. In the processing, the by-products are small broken pieces of low commercial value and are mostly sold for animal feed. However, they can also be transformed into highly valuable products like gelatin.

To use salted jellyfish by-products as raw material, the presence of remaining salt from the salting process of 20–25% [6, 7, 28] must first be removed. Therefore, there is a need for traditional desalination washing with water several times and overnight soaking before using it for food [6, 7, 12, 22]. This method requires time and is labor-intensive. A washing machine has been applied to reduce time, but the washing process generates a massive quantity of wastewater [28]. Research has proved that ultrasound can be used for protein desalination. Applying ultrasound and microwave pre-treatment in salted duck egg-white protein desalination can reduce salt content from 7.80 to 0.62% [29]. At present, ultrasound, a non-thermal process generating a sound wave at a frequency of 20–100 kHz, can be applied in food processing, such as in homogenization, emulsification, extraction, crystallization, degassing, marinating,

or cleaning. Ultrasound has been reviewed for pesticides, mycotoxins, heavy metals, and allergen removal [30]. The ultrasound frequency produces cavitation, crushing, vibration, mixing, and heating, thereby enhancing mass transfer to induce rapid bubble collapse and produce shear forces to break covalent bonds in the materials [31]. Applying this green technology, ultrasound could be an alternative method for eliminating the salts remaining in salted jellyfish by-products. However, no application of the ultrasonic method to desalinate such a jellyfish by-product sample has been reported.

Jellyfish gelatin research shows that the quality of gelatin gel is inferior to that of fish and bovine gelatin [6, 7, 10, 11, 32]. Comparing two species of jellyfish, the gel strength of gelatin gel from *Lobonema smithii* is higher than that from *Stomolophus meleagris* due to various factors related to the process, such as type of acid, extraction time, and temperature. However, the raw material preparatory step of *Stomolophus meleagris* displayed excessive salt content (or ash), close to 56.61% (dry weight basis), while the quantity of ash of *Lobonema smithii* after washing for 2 cycles decreased from 21.18 to 1.39% [28]. The ash content in dried powder was 15.76% [33]. Upon this research, the salt remaining in the raw material influencing the gelatin quality must, therefore, be further before gelatin extraction. The salts might reduce the strength of the gelatin network, thereby lowering gel strength. Therefore, the objectives of this study were to determine the combined effect of traditional washing subsequently to ultrasonic treatment on the reduction of the salt content of desalted jellyfish by-products and for gelatin production, to determine the effect of extraction time of combined wash-ultrasonicated desalted jellyfish on gelatin gel qualities.

## Materials and methods

### Preparation of ultrasonicated desalted jellyfish by-products

By-products of salted jellyfish (*Lobonema smithii*) were obtained from Mahachai Food and Trading Co. Ltd., Samutsakhon, Thailand. Minced salted jellyfish resulting from trimming was collected, packed in plastic bags (50 kg per bag), and transported to the Faculty of Applied Science, King Mongkut's University of Technology North Bangkok, Bangkok, Thailand. Given its excessive salt content, conventional washing was carried out. The jellyfish by-products were washed several times and soaked in tap water overnight. The resulting cleaned jellyfish by-products, with 0% measured by salinometer (Atago®, S/Mill-E, Japan), were stored at 4°C for a maximum of 2 weeks.

The conventional washed jellyfish samples were subjected to a laboratory sonication bath (model E 30 H, Elma-Hans Schmidbauer GmbH & Co KG Co. Ltd., Germany), working at a fixed frequency (40 kHz) and power (220 W) at a sample/water ratio of 1:5 (w/v). The sonication times varied from 20, 40, 60, and 80 to 100 min, designated as U20, U40, U60, U80, and U100. After ultrasound treatment, the samples were drained and dried at 60 °C using a tray drier (Dwyer, TDII, Thailand) for 24 h, ground, and sieved at 35 mesh. The moisture content of all dried ultrasonicated jellyfish is 9.94%. All jellyfish samples were kept in polyethylene bags at room temperature.

**Gelatin extraction.** Gelatin was then extracted from the ultrasound-treated jellyfish samples according to a method previously described, with slight modifications [10, 27]. Ultrasound-treated jellyfish samples (200 g each) treated for 0, 20, 40, 60, 80, and 100 min and designated as U0, U20, U40, U60, U80, and U100 were suspended in a 0.05 M NaOH solution in the ratio of 1:15 (w/v). The samples were stirred for 2 h continuously at 4 °C using a shaker (WiseCube, WIS-20R, Korea) at a speed of 150 rpm. After stirring, the samples were washed with tap water until the pH level of each sample was neutral (6.5–7.0). Then, an acid treatment was performed by soaking the sample in a 0.2 M HCl solution and stirring for 2 h at 25 °C.

After that, the residue was re-washed in tap water until a neutral pH was reached. Finally, the extraction was carried out at 80 ˚C at a 1:10 (w/v) ratio for 4, 6, and 8 h in a temperature-controlled shaker water bath (Memmert, Schwabach, Germany) at a speed of 70 rpm. The jellyfish gelatin solutions were then filtered using a Buchner funnel with Whatman filter paper No. 1 (Whatman International, Ltd., Maidstone, England), and the filtrates were subjected to a tray dryer (Dwyer, TDII, Thailand) at 60 ˚C. The resulting dried jellyfish gelatin sheets, with a moisture content of 6.8%, were milled and kept in a desiccator for further analysis. The gelatin samples, designated as JFG0, JFG20, JFG40, JFG60, JFG80, and JFG100, were then analyzed. The flow diagram of jellyfish gelatin extraction is shown in Fig 1.

## Analysis

**Sodium chloride measurement.** The salt content of the desalted jellyfish was determined following the AOAC method [34]. The minced desalted jellyfish (1g) by-products were added to an Erlenmeyer flask containing 25 mL of 0.1 M silver nitrate and 10 mL of concentrated nitric. The solution was mixed and boiled for 10 min. Then, 50 mL of distilled water and 5 mL of the ferric indicator were added to the solution. Finally, the solutions were titrated with 0.1 M potassium thiocyanate solution until a light brown solution was obtained. The salt content and the change in salt content (%) were calculated according to Eqs (1) and (2).

$$\text{Salt content (\%)} = 5.8 \text{ x} (V_1 \text{ x } N_1) - (V_2 \text{ x } N_2/W) \tag{1}$$

where $V_1$ is the volume of $AgNO_3$ (mL); $N_1$ is the concentration of $AgNO_3$ (N); $V_2$ is the volume of KSCN (mL); $N_2$ is the concentration of KSCN (N), and W is the weight of the sample (g).

$$\text{Salt reduction (\%)} = (S_0 - S_i)/S_0 * 100 \tag{2}$$

where $S_0$ is the salt content before sonication and $S_i$ is the salt content after sonication.

**Water absorption.** Water absorption is the amount of water absorbed by a sonicated jellyfish sample and is calculated as a ratio of the weight of water absorbed after sonication. Each sample of traditionally washed jellyfish by-products used in each sonication weighed 200 g originally. After sonication, the sample was drained for 5 min and then weighed again. The water absorption was calculated according to Eq (3).

$$\text{Water absorption(\%)} = (W_i - W_0)/W_0 * 100 \tag{3}$$

where $W_i$ is the drained weight of each sonicated sample; i = 0,20, 40, 60, 80, or 100, and $W_0$ = the initial weight of the sample with no sonication (200 g).

**Gelatin yield.** For each gelatin extraction, the dried sonicated jellyfish of 20 g were used. After extraction and drying, each jellyfish's gelatin having a moisture content of 6.8%, was weighed. The yield was calculated according to Eq (4).

$$\text{Yield (\%)} = \text{weight of dried gelatin obtained (g)}/20 \text{ g x } 100 \tag{4}$$

**Moisture content.** The moisture content of samples (dried ultrasonicated jellyfish samples or dried jellyfish gelatins) was determined according to AOAC [34] by drying the sample in an oven at 105˚C until constant weight. The dried sample was weighed and then calculated according to Eq (5).

$$\text{Moisture content (\%) dry weight basis} = (W_s - W_d)/W_d * 100 \tag{5}$$

For dried ultrasonicated jellyfish samples, Ws is the initial weight of the sample (200g), and

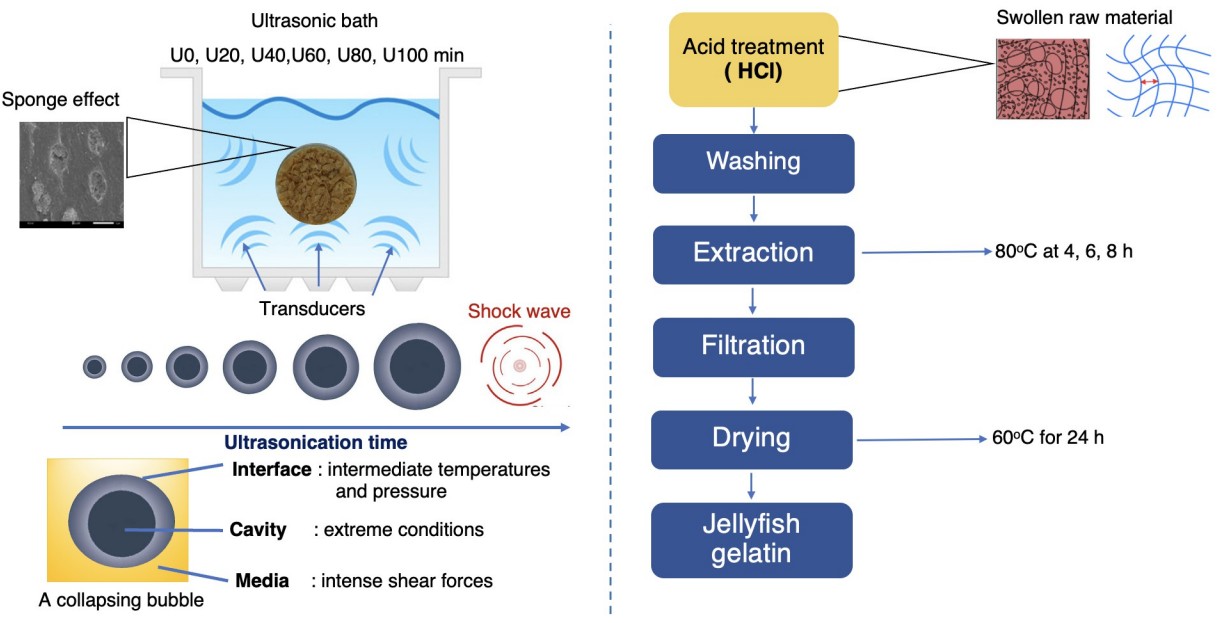

**Fig 1. The sonication assisted in washing salted jellyfish by-products and the jellyfish gelatin extraction process.**

Wd is the weight of the dried sonicated sample. For the calculation of dried jellyfish gelatin, Ws is the initial weight of the sample (20 g), and Wd is the weight of the dried jellyfish gelatin sample.

**Soluble protein determination.** After sonication of desalted jellyfish by-products, the soluble jellyfish protein in a water medium was measured by the Lowry method with BSA as the standard protein [35].

**Conductivity measurement.** The water medium's electrical conductivity (EC) in the sonication bath was determined directly using a conductivity meter (Starter 3100, Starter Bench Lab Meter, USA).

**Microstructure analysis.** The microstructure of ultrasound-treated jellyfish raw materials and all jellyfish gelatin gels was visualized using scanning electron microscopy (JEOL JSM-6610LV, Tokyo, Japan) with acceleration voltage at 15 kV. For the surface morphology, each of the dried sonicated jellyfish samples was coated using a gold-palladium alloy coater (JFC-1200 fine coater, JEOL, Japan). The microstructure of jellyfish gelatin gel samples consisting of 6.67% gelatin was prepared following a slightly modified version of the method described by Sae-leaw et al. [36]. The cross-section of ultrasound-treated jellyfish raw materials was magnified and observed at 1,000 and 5,000 times, while those of gelatin gels were observed at 10,000 and 30,000 times.

**Fourier transforms infrared (FTIR) spectroscopic analysis.** The infrared spectra of ultrasound-treated jellyfish raw materials and all jellyfish gelatin powder samples were analyzed by FTIR spectrometer (Jasco Inc., Easton, MO). The dried ultrasonicated jellyfish by-products and freeze-dried jellyfish gelatins were used for FITR analysis. The samples were ground and mixed with potassium bromide (KBr) to form thin pellets using an MP-1 hydraulic press and pressed under 0.9 MPa for 4 min to produce 13 mm diameter pellets (JASCO Corporation, Tokyo, Japan). The pure KBr pellets were used for background measurement. The analysis was performed at ambient temperature using a scan range of 400–4000 cm$^{-1}$ at a resolution of 4 cm$^{-1}$ [10].

**Gel strength.** The gel strength of jellyfish gelatin samples was determined according to the Gelatin Manufacturers Institute of America (GMIA) with a slight modification [10]. The jellyfish gelatin gel was prepared with a 6.67% gelatin solution for all measurements. Samples were transferred to a setting mold with dimensions of 3 cm diameter and 2.5 cm height and incubated at a refrigerated temperature (4 ˚C) for 18 h. Gel strength was determined using a texture analyzer (Stable Micro System, Surrey, UK).

**Thermal properties.** The gelling and melting temperatures of gelatin gel prepared at 6.67% solution were investigated by a small deformation oscillatory measurement using a controlled stress rheometer (Gemini 200 HR Nano, Malvern Instruments, UK). The parallel plate of 40 mm diameter with a 0.5 mm gap in oscillatory mode was set up. A temperature sweep was carried out from 40 ˚C to 5 ˚C and 5 ˚C to 40 ˚C, with a constant heating and cooling rate of 2 ˚C/min. The gelatin solution was poured into the parallel plate and covered with paraffin oil [10].

**Preparation of jellyfish gelatin peptide and identification by LC-MS/MS.** The experiment was divided into two sections. The first section was sample digestion with trypsin and the second section was protein peptide identification by LC-MS/MS. Before measurement, all bovine, fish, and jellyfish gelatin samples of JFG40-4 (highest gel strength) and JFG100-8 (lowest gel strength) were desalted by dialysis against distilled water overnight. Each sample was firstly digested with trypsin and the peptides were then identified by LC-MS/MS. The gelatin solution was solubilized in 10 μL of sample buffer (10% SDS, 10 mM dithiothreitol (DTT), 1.5 mM Tris-HCl, pH 8.8) and centrifuged at 10,000 g for 10min. The supernatant was mixed with 50 μL of 10 mM DTT in 10 mM $NH_4HCO_3$ (Sigma-Aldrich) and incubated at 56 ˚C for 60 min. After that, 50 μL of 100 mM iodoacetic acid (IAA) (Sigma-Aldrich) in 10 mM $NH_4HCO_3$ was added and further incubated in the dark at room temperature for 60 min. The sample was diluted with 40 μL of 25 mM $NH_4HCO_3$ containing 2μg of trypsin. After overnight incubation at 37 ˚C, the peptides were collected and dried with a speed vacuum concentrator (Thermo Scientific, USA).

The peptide analysis of the samples was performed using a Hybrid quadrupole Q-TOF impact II™ (Bruker Daltonics, Germany) coupled with an Ultimate3000 Nano/Capillary LC System (Thermo Scientific, USA). The dried sample was re-dissolved in 0.1% formic acid. The mixture was loaded onto a C18 reversed-phase column having a length of 15 cm and a diameter of 75 μm packed with an Acclaim PepMap® RSLC C18, 3 μm, 100Å, nanoViper (Thermo Scientific, USA). After equilibration with solvent A (0.05% Trifluoroacetic acid in water), the tryptic peptides were separated with a linear gradient of 5–55% solvent B (0.1% formic acid in 80% acetonitrile) at a flow rate of 0.3 μl/min over 45 min. Electrospray ionization was carried out at 1.6kV using the CaptiveSpray. Nitrogen was used as a drying gas (flow rate about 50 l/h). Collision-induced-dissociation (CID) product ion mass spectra were obtained using nitrogen gas as the collision gas. Mass spectra (MS) and MS/MS spectra were obtained in the positive-ion mode at *2 Hz* over the range (*m/z*) 150–2200 (Compass 1.9 software, Bruker Daltonics). The collision energy was adjusted to 10 eV as a function of the *m/z* value. The MaxQuant (version 1.6.6.0) was used to quantify individual samples bioinformatically, and their MS/MS spectra were matched to the UniProt database using the Andromeda search engine [37]. The protein sequences were searched in FASTA files. The protein database was downloaded from the UniProt database of fish gelatin, bovine gelatin, bovine collagen, and alpha 2 and 1 chain. The MaxQuant ProteinGroups.txt file was loaded into Perseus version 1.6.6.0, and potential contaminants that did not correspond to any UPS1 protein were removed from the data set [38]. Max intensities were log2 transformed, and pairwise comparisons between conditions were made via t-tests. Perseus also imputed missing values using a constant value (zero). The visualization and statistical analyses were conducted using a MultiExperiment

Viewer (MeV) in the TM4 suite software [10]. The Venn diagram displays the differences between protein lists originating from different samples.

**Statistical analysis.** Each experiment was performed in triplicate. The data were subjected to analysis of variance (ANOVA). Duncan's multiple range test was used to compare the mean value. Statistical analysis was performed using the Statistical Package for Social Science (SPSS 17.0 for windows, SPSS Inc., Chicago, IL, USA).

# Results and discussion

## Quality of ultrasonicated desalted jellyfish by-products

The salt content of each condition (U0, U20, U40, U60, U80, and U100) was initially measured (S1 Table in S1 File), and the salt reduction compared to the control (U0) was then calculated. S2 Table in S1 File shows the quality of ultrasonicated jellyfish by-products. The salt content of traditionally washed jellyfish subjected to ultrasonic waves was significantly reduced with an extended exposure time of the ultrasound. The salt reduction was significantly increased after the first 20 min of exposure, and the total salt content was removed when the sample was sonicated for 100 min. When a sample is placed in the sonication bath, ultrasonic waves having high energy generate cavities that are transported in the medium and interact with the protein's surface [30]. As a result, the cavities collapse and release energy that facilitates the detachment of salts on the protein surface, thereby reducing the salt content of the by-products [39, 40]. In addition, the ultrasonic waves affected the degradation of jellyfish protein. The extended ultrasound time increases the temperature of the water medium by 1 ˚C at every 5 min of exposure time. The starting temperature of the water medium was 25 ± 3 ˚C (ambient temperature). After 100 min of sonication, the temperature of the water medium increased to 45–54 ˚C, thereby resulting in increased soluble protein and salt in the water medium. The effect and energy waves cause the degradation of minced desalted jellyfish protein by-products. Apart from increased soluble jellyfish protein in the water medium, the increased water medium conductivity by increased salt solubility was also found. The result indicates that the ultrasound technique could be a method of choice for removing salt from jellyfish by-products. The results were similar to the results of the prominent desalination rate of salted egg white by ultrasonic waves [29].

The ultrasound effect on jellyfish protein showed increased water absorption when the sample was exposed to ultrasonic waves for a short period of 20 and 40 min. After that, the decreased water absorption was remarkably found at 80 and 100 min of ultrasound exposure (S1 Table in S1 File). When the minced by-products were used, the long ultrasonic exposure time caused shrinkage on texture and generated more broken pieces by the action of energy waves that enhances the transport of water molecules from the collagen structure of jellyfish, resulting in the dense collagen helix and shrinkage appearance. The previous report found a reduction of 25–30% of the initial length of jellyfish collagen if the jellyfish protein was subjected to a denaturation temperature of 60 ˚C [40]. The measured color was expressed as L* (lightness), a* (red-green), and b* (yellow-blue). In this study, the jellyfish by-products appeared to be broken pieces with brown color. When the jellyfish samples were exposed to ultrasonic waves, the appearance and color slightly changed. The sonicated jellyfish by-products showed a decreased L* value but increased a* and b* values after extended ultrasound exposure. The ultrasonic wave facilitates the penetration of water molecules into the collagen matrix and flows out of the matrix, thereby generating the shrinkage of jellyfish collagen [39]. In addition, changes in the color and texture of jellyfish by-products are caused by the ultrasound effect.

## Microstructure of ultrasonicated jellyfish by-products

The action of ultrasonic waves in washing facilitates the removal of salts from the collagen surface of desalted jellyfish. The jellyfish by-product samples used in this experiment were conventionally washed. As seen in Fig 2, on the surface of jellyfish collagen without ultrasonic exposure, salt crystals remained on the collagen surface due to the bonding of negative or positive charges of protein, as well as ionic bonding within salt molecules. Conventional washing or machine washing cannot eliminate all of the salt from jellyfish samples. The microstructure results of jellyfish by-products revealed the scar remaining on the jellyfish sample surface caused by the direct crystal salts added in the salting process. The effect of ultrasound frequency generates vibration waves that travel from the water to the protein surface and cause the detachment of salts from the surface of the jellyfish protein. The longer the ultrasound time, the greater the removal of salts from the protein (S2 Table in S1 File). However, no sophisticated analysis of elements by X-ray fluorescence spectroscopy was applied in this study. The previous research reported that washed jellyfish by a washing machine for 30 min (2 cycles X 15 min)showed the remaining elements of Na, Al, Si, P, S, Cl, K, Ca, Fe, and Br valued at 11.65, 7.87, 1.32, 1.59, 9.17, 65.14, 1.46, 1.29. 0.21 and 0.25%, determined by X-ray fluorescence spectroscopy [28]. The application of ultrasonic waves to the washed jellyfish by-products effectively causes the removal of salts from the surface of jellyfish protein (Fig 2.). When the extended ultrasonic time was applied, the salts in the jellyfish by-products dissolved, thereby increasing conductivity. The results correlate with the salt reduction after ultrasound treatment in S2 Table in S1 File. During the salting process, the consequence of adding a high quantity of dry salts shows denatured protein in the form of scars on the surface of collagen protein. Therefore, using ultrasound to eliminate salts in the washing process provides benefits by shortening the washing time to approximately 12 h and removing salts in the sample. The salt removal from salted jellyfish is a crucial step in using jellyfish in food dishes and for gelatin

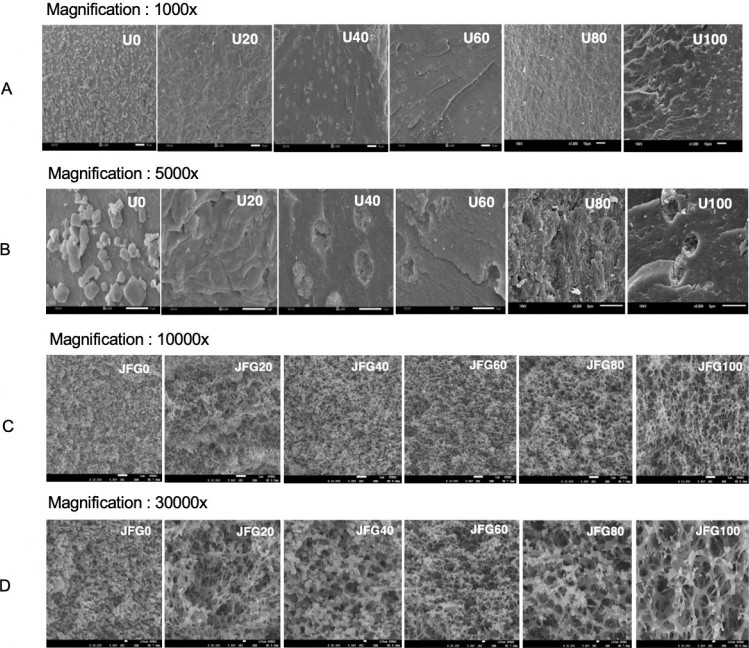

**Fig 2. Scanning electron micrographs of ultrasound-treated jellyfish by-products (A-B) and ultrasonic-treated jellyfish gelatin (C-D).**

research. The drawback of the high salt (sodium) in food products is that it can lead to higher blood pressure and a significant risk of heart disease and stroke. The excessive salt in the raw material used for gelatin production also reduces gelatin gel yield and gel strength [32].

## Infrared spectroscopic analysis of ultrasonicated jellyfish powder and jellyfish gelatin

Infrared spectroscopic analysis of ultrasonicated jellyfish powder and jellyfish gelatin monitored the chemical structure of intact collagen. The spectra of jellyfish powder showed similar characteristic spectral bands (amide A, B, I, II, III) of collagen. The N-H stretching vibration of the amide A band was observed at 3438.57, 3400.82, 3432.83, 3433.65, 3424.62, and 3436.11 cm$^{-1}$ for treated with ultrasonic waves from 0, 20, 40, 60, 80, and 100 min, respectively. Amide B bands appeared at 2931.32, 2928.86, 2926.40, 2925.74, 2923.93, and 2920.65 cm$^{-1}$, corresponding to C-H anti-symmetrical stretching. The change of collagen secondary structure and hydrogen bonding between N-H stretch and C = O represent the amide I, II, and III at around 1660, 1550, and 1220 cm$^{-1}$, respectively. The shift of the amide I band's wavenumbers dictates the triple helix's significant loss via the breaking down of H-bonds between α-chains [41], thereby changing the secondary structure of jellyfish collagen. Amide, I band from FTIR spectra was slightly shifted, as shown in Fig 3A. It indicates that ultrasonication does not affect the crosslink between protein molecules. The change of amide II also dictates the secondary structure changes of collagen and the hydration of protein [42]. The amide II vibration mode is attributed to the peptide group's out-of-phase C-N stretch and in-plane N-H deformation modes. Bands of amide II were slightly decreased and observed at 1550.75, 1547.46, 1545.00, 1542.54, 1539.26, and 1536.79 cm$^{-1}$ for treated with ultrasonic waves from 0, 20, 40, 60, 80, and 100 min, respectively. The amide III represents the combination peaks between C-N stretching vibration and N-H deformation from amide linkages and absorptions arising from wagging vibrations from CH$_2$ groups of the glycine backbone proline sidechains, which are involved in the intermolecular interactions of collagen [43]. The amide III bands of the exposed samples

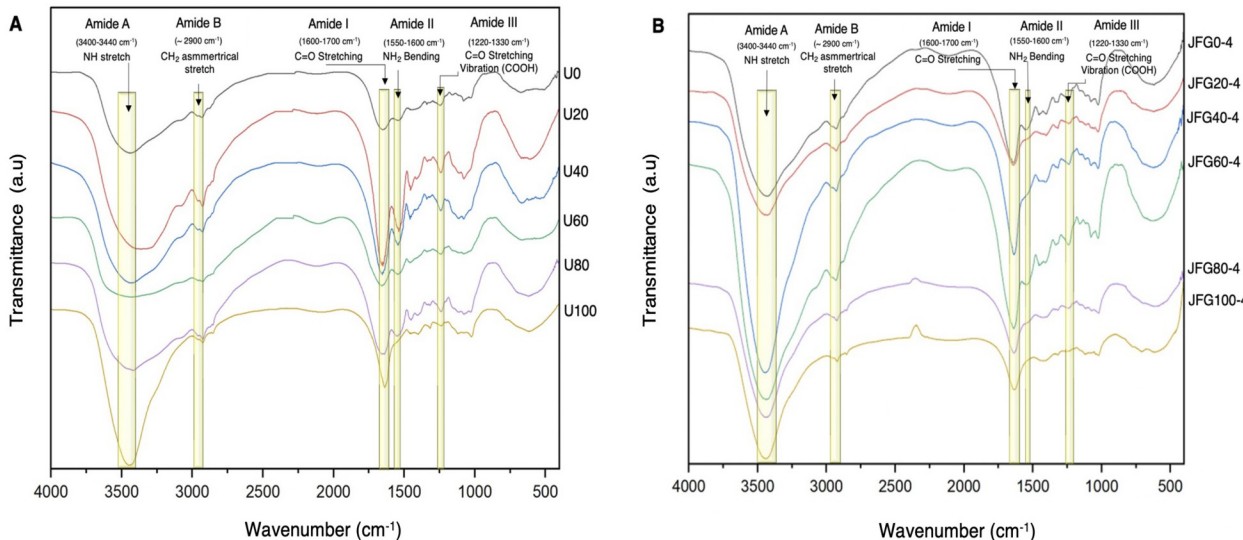

**Fig 3. Fourier transform infrared spectra of ultrasound-treated jellyfish by-products (A) and jellyfish gelatins extracted from different ultrasonicated jellyfish by-products (B).** A: The ultrasound settings used were 40 kHz and 220 W. The duration time of ultrasound treatment was 0, 20, 40, 60, 80, and 100 min, designated as U0, U20, U40, U60, U80, and U100, respectively. B: The jellyfish gelatins extracted from different ultrasonicated jellyfish by-products are designated as JFG0, JFG20, JFG40, JFG60, JFG80, and JFG100, respectively.

to ultrasonic waves from 0, 20, 40, 60, 80, and 80 min appeared at 1244.59, 1243.77, 1241.31, 1238.85, 1236.38, and 1228.18 cm$^{-1}$, respectively. Although, the wavenumber of characteristic spectral bands collagen (amide A, B, I, II, III) was shifted to lower frequencies in 100 mins ultrasonically treated sample. The longer ultrasonication duration mighthe t increase the thermal and unstable crosslink of protein molecules. However, a slight difference in the wavenumber of amide II and III can indicate that collagen structure might remain the same as in the untreated. Apart from the effect of ultrasonic waves on jellyfish collagen, the desalination effect of salted jellyfish by ultrasound also lowered the wavenumbers of the amide III band. Nadzrin et al. [44] reported the effect of salt content on the amide III band. The H$^+$ of salt content can facilitate complexation with atoms in the amide III functional group, increasing ionic conductivities and lower wavenumber.

The gelatin was extracted from ultrasonicated jellyfish by-products at 80°C for 4 h. The FTIR spectra of gelatin are shown in Fig 3B and S3 Table in S1 File. All jellyfish gelatins exhibit major absorption bands in the amide band region. The spectrum of amide I peak that appears at 1600–1700 cm$^{-1}$ is mainly related to secondary [45, 46]. The amide I band of JFG0-4 to JFG100-4 was shown at 1649.75, 1643.29, 1643.06, 1636.47, 1631.93, and 1629.65 cm$^{-1}$, respectively. The high-temperature extraction might cause losses of molecular order of triple helix of collagen due to thermal uncoupling of inter-molecular crosslink. The collagen denaturation was interpreted as in disintegration of the triple helix collagen structure into a random coil [43].

The amide II bands were found at a spectrum between 1550–1600 cm$^{-1}$. All the samples had a band with insignificant differences in the spectra of the amide II band. They were found at the wavenumbers of 1550.75, 1547.46, 1545.00, 1542.54, 1539.26, and 1536.79 cm$^{-1}$, respectively. Amide II bands arise from the in-plane bending mode of NH of the peptide groups and the stretching vibration of CN groups [45]. The lower wavenumber of JFG100-4 indicated more intensity of NH involved in hydrogen bonding with adjoining alpha chains [46]. The amide III region occurs in the range of 1220–1330 cm$^{-1}$. The band at 1245–1270 cm$^{-1}$ is attributed to random coil structure [47]. The vibration peak at near 1270 cm$^{-1}$ revealed the amide III bands of collagen, which is related to the triple helical structure of collagen [48]. The change in a band indicates the unstable helical structure, which affects weak gel properties. Prior to extraction, the ultrasonic wave treatment primarily caused damage to the muscular structure and induced disruption of the helical structure of ultrasonicated jellyfish by-products [45]. When subjected to high temperature, the molecular change from an α-helical structure to a random coil, showing collagen denaturation to gelatin [49]. Moreover, the reduction in the amides I and III band intensity by extraction time was associated with the more significant loss of molecular order in jellyfish gelatin. The collapsing cavitation bubbles lead to increased surface area and solvent absorption in the jellyfish tissue layer, enhancing the accessibility of the water as solvent resulting in increased extraction yield, which, in turn, lowers the gel strength.

The spectral region between 3400–3440 cm$^{-1}$ corresponds to the band of amide A, whereas amide B corresponds to the asymmetric stretching vibration of = C-H and -NH$^{3+}$ [5] was observed at 2931–2921.85 cm$^{-1}$. The band shift of amide A and amide B to lower wavenumbers suggests a hydrogen bond's sensitivity with high-temperature extraction and the effect of high-temperature facilitating the interactions of -NH$^{3+}$ groups between peptide chains.

The long extraction time of 8 h (JFG0-8) showed that all amides A, B, I, II, and III had lower wavenumbers than 4 and 6 h, thereby changing the conformation of jellyfish gelatin from helix to random coil. The possibility of the thermal uncoupling effect and the protein aggregation during extraction time might also have occurred.

## Quality of jellyfish gelatin extracted from ultrasonicated by-products

**Jellyfish gelatin yield.** The yield of jellyfish gelatins produced from ultrasonicated jellyfish by-products with different exposure times of ultrasound (0, 20, 40, 60, 80, 100 min) and different extraction times (4, 6, 8 h) had a different yield varied from 7.43 to 32.69%. Using the same ultrasonicated sample, the yield of jellyfish gelatin extracted for 8 h showed higher values than those extracted for 4 and 6 h. The highest yield was 32.69 ± 1.98%, derived from the ultrasonicated jellyfish for 80 min and then extracted for 8 h; JFG80-8H, and the lowest yield was from the sample with no ultrasound treatment (S4 Table in S1 File). The effect of ultrasonic waves on the yield of 7.43±0.15% was from the sample with no ultrasound treatment, JFG0-4H (S4 Table in S1 File). The effect of ultrasonic waves on the yield of gelatin results in the generation of acoustic cavitation bubbles of high energy that collapse hydrogen bonds and decrease peptide bonds' stability in the collagen structure of the jellyfish sample. In addition, the bubbles facilitate the increased dissolution area, and Cavitation is enhanced at a long extraction time, but the resulting ultrasonicate effect facilitates the loosening of pre-treated jellyfish collagen. Subsequently, hot water extraction loosens and destabilizes the collagen structure, thus inducing the helix-to-coil transition of collagen to soluble gelatin [49, 50]. The use of ultrasound was also found in poultry and marine gelatin. Duck feet gelatin extracted in a sonication bath at 40 kHz, 60˚C for 10 min showed a high yield, affected by the combination of high temperature and pressure [50]. The positive effect of ultrasound on yield was also reported in clown featherback (*Chitala ornata*) skin gelatin, with gelatin yield increasing from 23.46% to 57.35% by ultrasound pre-treatment at 80% amplitude for 30 min [45]. However, a higher gelatin yield was obtained by increased duration time of the ultrasound treatment [51] to form gelatin [28]. When a high temperature is applied during gelatin extraction, the conversion of collagen to gelatin is caused by the destruction of the stabilizing hydrogen bonds of collagen, resulting in the transformation of the helix-to-coil structure to a randomly disorganized structure [28, 32]. Therefore, the effect of ultrasound from the preparatory step of jellyfish by-products and the short duration time of hot water extraction offers a relatively high yield of gelatin production.

**Gel strength and viscosity of jellyfish gelatin.** The gel strength of gelatin dictates gel quality for potential use in food products. The synergistic effect of ultrasonicated jellyfish samples and extraction time delivered different gel strengths in jellyfish gelatin. Results showed that in the hot water extraction process at 80˚C, jellyfish gelatin gel extracted at 4 h had higher gel strength values than gel extracted at 6 and 8 h, regardless of the preparative time of sonication. Compared to all samples of ultrasonicated jellyfish by-products used. The gel strengths of gelatin extracted for 4h of JFG0-4, JFG20-4, JFG40-4, JFG60-4, JFG80-4, and JFG100-4 were 129.62 ± 1.10, 160.05 ± 1.63, 447.01 ± 1.06, 283.29 ± 2.36, 278.71 ± 2.90 and 121.66 ± 2.27 g, respectively. The JFG0-6, JFG20-6, JFG40-6, JFG60-6, JFG80-6, and JFG100-6 for extraction time 6 h were 188.70 ± 1.79, 246.76 ± 1.98, 278.91 ± 1.59, 191.98 ± 3.04, 150.33 ± 2.95 and 120.06 ± 1.10 g, respectively. For 8 h of extraction, the gelatin gel strength samples of JFG0-8, JFG20-8, JFG40-8, JFG60-8, JFG80-8, and JFG100-8 were 248.12 ± 3.60, 207.75 ± 2.55, 181.57 ± 1.42, 152.95 ± 2.88, 114.92 ± 1.85, and 91.89 ± 1.74 g, respectively. Jellyfish gelatin gel produced from by-products sonicated for 40 min and extracted for 4 h yielded the highest gel strength of 447 ± 1.06 g (S4 Table in S1 File). The highest gel strength of jellyfish gelatin has been reported for the first time compared to, in this study, the bovine and fish gelatins had gel strengths of 540.06 g and 640.65 g. The main difference between marine gelatin and mammalian gelatins is in the imino acid content, where the imino acids of mammalian gelatins were reportedly higher than marine gelatins [5]. Imino acids; hydroxyproline, and proline, play a role in forming a gel. The hydroxyl groups help to stabilize the collagen helix by hydrogen bonding to the water molecules and direct hydrogen bonding to the carbonyl group [36].

However, the effect on ultrasonicated desalted jellyfish and hot water extraction for a long time generates the broken minced collagen of by-products, thereby producing shorter chain lengths of peptides that lower inter-junction zones of gelatin gel [39].

Viscosity is the second most important physical property of gelatin after gel strength. Generally, a gelatin solution with low viscosity usually yields a short and brittle textured gel, while a highly viscous gelatin solution yields a tough and extensible gel [51]. The viscosity of all jellyfish gelatin samples was 7cP, but the commercial gelatins reported were in the range of 15–28 cP. The lower viscosity of jellyfish gelatin might be due to peptide chains with low molecular weight, resulting from the shortened collagen structure during the pre-treatment step.

*Thermal properties of jellyfish gelatin.* The heating and cooling curve (melting point) shows the storage modulus (G') during the gel formation of jellyfish gelatin, thereby presentingmelting the point of gelatins. The steepness of G' was observed when the temperature was changed from 5 to 40 ˚C. The melting point of commercial gelatin compared with the jellyfish gelatin produced from by-products with different ultrasonicate times and extracted at 80˚C for 4 h was in the range of 13–21 ˚C. The highest melting point of 21 ˚C was from samples treated with ultrasound for 40 and 60 min. Compared to fish and bovine gelatin, all jellyfish gelatins melted easily at room temperature. The gelatin gels are formed through the transition of gelatin chains from single-strand to triple-helix via ionic interaction, hydrogen bonding, van der Waals forces, hydrophobic association, and self-assembly [51]. G' decreased as the temperature increased during the heating stage, indicating that the gelatin gel network was weakening. Due to the inferior gel property of jellyfish gelatin, combining jellyfish gelatin and other hydrocolloids might help to improve the melting temperature and benefit applications in food and pharmaceutical products [50].

*Identification of jellyfish gelatin peptides.* Certain peptides can be used to dictate the gel properties of jellyfish gelatin. However, some studies have shown the results of the identified peptides using a proteomic approach. Different collagen types and amino acid sequences might explain the differences in the textural characteristics of gel from jellyfish gelatin produced from by-products with different ultrasonicated times. Fig 4 shows the Venn diagram of characteristic tryptic peptides of jellyfish from gelatin gels extracted at 4 and 8 h with the highest and the lowest gel strengths (JFG40-4 and JFG100-8), fish gelatin (FG), and bovine gelatin (BG). Fig 5 shows the abundance of each protein found in the gelatin samples. Table 1 displays the proteins and amino acid sequences found in the gelatin samples. Results show that BG, FG, JFG40-4, and JFG100-8 had identified peptides of 19, 27, 24, and 24. The 21 peptides were similarly found in JFG40, FG, and BG. Compared to the same marine gelatin, 19 proteins in JFG40-4 and FG were closely related. In addition, 17 proteins were found in both JFG40-4 and JFG100-8. The collagen alpha 2 (I) was present in all the gelatin and had sequences of GDGAPGHPGPPGAPGVEGKDGPPR. The JFG100-8 lacked the amount of protein that the JFG40-4 and commercial gelatin (FG>BG>JFG40-4>JFG100-8). Glycine is the most abundant amino acid in type I collagen, and all the type I collagen was rich in Ala, Pro, Hyp, and Glu [43]. Lueyot et al. [10] reported that 29 unique peptides were obtained in bovine, fish, and jellyfish gelatin to confirm the effect on gel properties. The 29 proteins commonly found in jellyfish gelatin lack abundance more than BG and FG, providing low gel properties in jellyfish gelatin. The unique peptide sequence of the collagen alpha-2(I) chain and collagen alpha-2(IV) chain influenced gel properties, bovine and fish gelatin had the sequences of AGPPGPPRGA GAP GQSFLLR and AEQGEFYLLSYGSWKLNMGVPCMPEQDTQS, respectively.

Interestingly, collagen alpha-2 type VIII was found only in BG. The differences between BG and FG were 11 proteins: collagen alpha-1 type XXVI, collagen alpha-1 type III, collagen alpha-2 type IV, collagen alpha-3 type IV, collagen alpha-1 type VII, collagen alpha-1 type XVII, collagen alpha-1 type XXI, collagen alpha-1 type XVI, collagen alpha-3 type VI, collagen

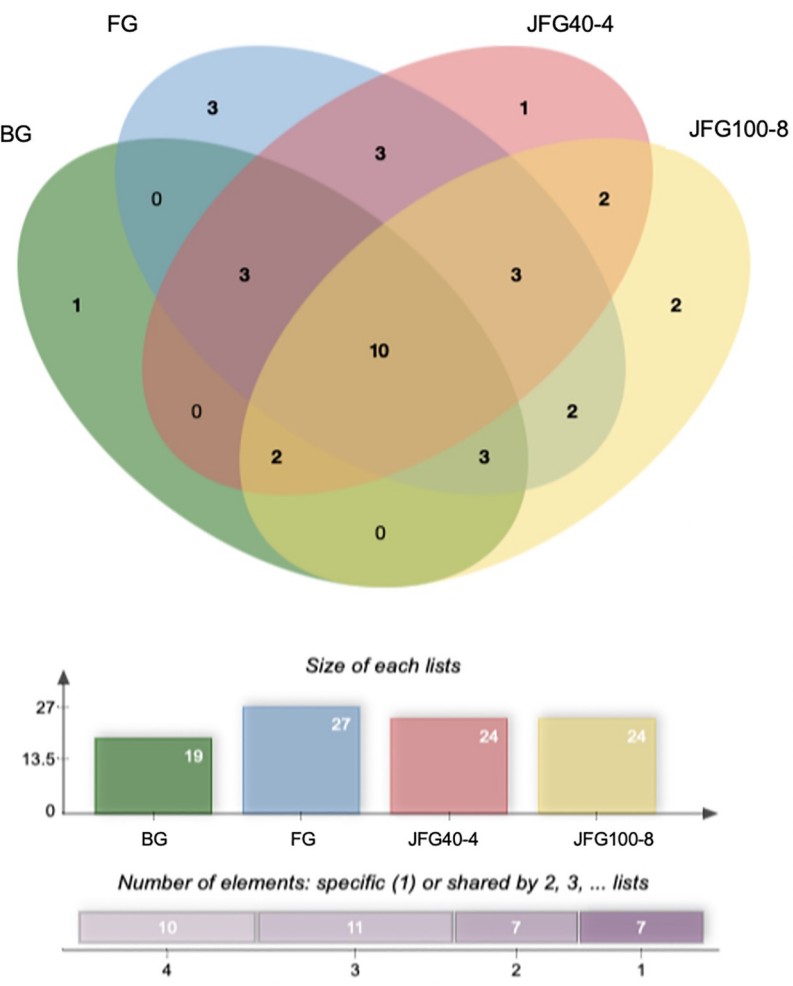

**Fig 4. Venn diagram of tryptic peptides of jellyfish, fish, and bovine gelatin.**

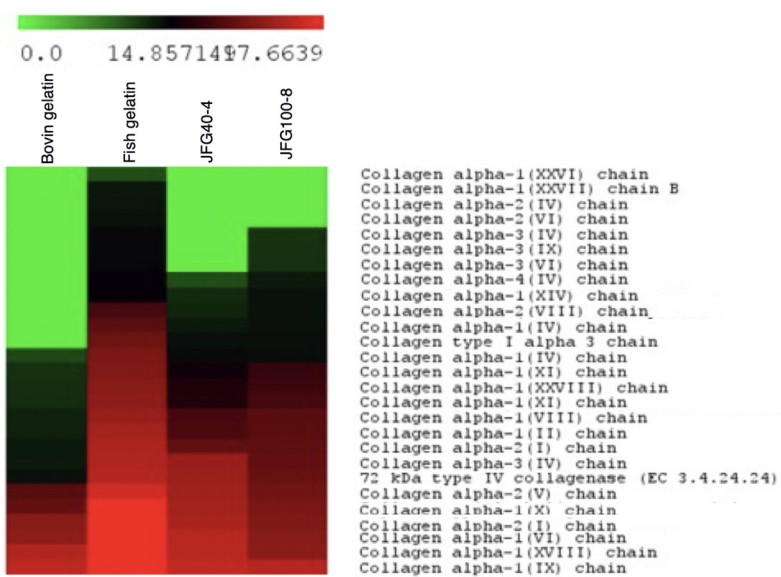

**Fig 5. Heatmap showing the abundance of each protein in jellyfish, fish, and bovine gelatin.**

**Table 1. Protein name and sequence of bovine (BG), fish gelatin (FG), and jellyfish gelatin (JFG40-4 and JFG100-8) produced from different ultrasonicated jellyfish by-products.**

| Protein names | Sequence | BG | FG | JFG 40 | JFG 100 |
|---|---|---|---|---|---|
| Collagen alpha-2(VIII) chain | GDQGPSGLAGKPGFPGDK | ✓ | | | |
| Collagen alpha-1(XXVI) chain | EGLLGKPGDAGPK | | ✓ | | |
| Collagen alpha-1(III) chain | GENGVPGEDGAPGPMGPRGAPGER | | ✓ | | |
| Collagen alpha-2(IV) chain | CSIYRPIDEEAVPMMPLPATTSLWYK | | ✓ | | |
| Collagen alpha-1(XV) chain | AFVSSHRQDLVHVVFPGFRETLPVTNLR | | | ✓ | |
| Collagen alpha-1(XXV) chain | IELLEMER | | | | ✓ |
| Collagen alpha-1(XXVII) chain B | CFVVGCKSEHK | | | | ✓ |
| Collagen alpha-1(XIV) chain | AYPPQNYQPSYPDPHASRLGPAHPGDTVGMR | ✓ | ✓ | ✓ | ✓ |
| Collagen alpha-1(IV) chain | GCLGVVGRQGNPGEPGEK | ✓ | ✓ | ✓ | ✓ |
| Collagen alpha-1(XI) chain | EGQSGEKGSLGPPGPQGPIGYPGPR | ✓ | ✓ | ✓ | ✓ |
| Collagen alpha-1(XXVIII) chain | DWRGVPTFQDR | ✓ | ✓ | ✓ | ✓ |
| Collagen alpha-1(II) chain | DLRDYRGCLER | ✓ | ✓ | ✓ | ✓ |
| Collagen alpha-2(I) chain | DAHLPIRTWLNDLGNNSK | ✓ | ✓ | ✓ | ✓ |
| 72 kDa type IV collagenase | AAFADDVTQPDLDPTYGIRIYTIQK | ✓ | ✓ | ✓ | ✓ |
| Collagen alpha-1(X) chain | GATGATGNKGDAGNTGAPGTPGAPGPAGPKGLQGYPGAAGEK | ✓ | ✓ | ✓ | ✓ |
| Collagen alpha-2(I) chain | GDGAPGHPGPPGAPGVEGKDGPPR | ✓ | ✓ | ✓ | ✓ |
| Collagen alpha-1(XVIII) chain | DKIGDNFIADWNDFLVEKDLNVIPLHR | ✓ | ✓ | ✓ | ✓ |
| Collagen alpha-1(VIII) chain | EMPHMPYGNEMPLLPQYGKERPQIPMHMGK | ✓ | ✓ | ✓ | |
| Collagen alpha-2(V) chain | FKHPSDITMVK | ✓ | ✓ | ✓ | |
| Collagen alpha-1(IX) chain | GEPGPVGPLGQKGSRGLR | ✓ | ✓ | ✓ | |
| Collagen alpha-1(IV) chain | GDIGPAGQPGPR | ✓ | ✓ | | ✓ |
| Collagen alpha-3(IV) chain | CPSGWLPLWQGYSFVMQTGAGAEGSGQPLVSPGSCLQEFRK | ✓ | ✓ | | ✓ |
| Collagen alpha-1(VI) chain | AGAPGYRGDEGPAGPEGGK | ✓ | ✓ | | ✓ |
| Collagen alpha-3(IV) chain | AGELENIISRCQVCMK | | ✓ | ✓ | |
| Collagen alpha-1(VII) chain | ARLAPEQQLAVKGEEGR | | ✓ | ✓ | |
| Collagen alpha-1(XVII) chain | DEVRQYLIGPPGPPGPPGVPGGYGFNTQEVAGR | | ✓ | ✓ | |
| Collagen alpha-1(XXI) chain | EGPPGPDGKPGPPGSR | | ✓ | | ✓ |
| Collagen alpha-1(XVI) chain | LLIVPRGPR | | ✓ | | ✓ |
| Collagen alpha-3(VI) chain | ATPMTPTVTVVEALSVDDSK | | ✓ | ✓ | ✓ |
| Collagen alpha-1(I) chain | EGQKGNRGETGAAGR | | ✓ | ✓ | ✓ |
| Collagen alpha-4(IV) chain | CAVCEAPAQAVAVHSQDQSIPPCPR | | ✓ | ✓ | ✓ |
| Collagen type I alpha 3 chain | DGPAGVSGPAGGPR | ✓ | | ✓ | ✓ |
| Collagen alpha-1(XI) chain | ASRTLAMPCPSTPWAPPGFYFPLALPLSSLPLTPPR | ✓ | | ✓ | ✓ |
| Collagen alpha-3(IX) chain | DGEKGSR | | | ✓ | ✓ |
| Collagen alpha-2(VI) chain | FADLVAEDFIDRIENVLCPEPAVANCIQR | | | ✓ | ✓ |

alpha-1 type I, and collagen alpha-4 type IV. Of the 27 proteins in FG, 19 proteins were also found in JFG40-4. In addition, 3 proteins (collagen alpha-1 type XV, collagen alpha-3 type IX, and collagen alpha-2 type VI) were found in JFG40-4. The difference in ultrasound pretreatment and extraction time could cause a change in gelatin polypeptides and have a shorter triple-helical domain of type VI, X, and XIV type I collagen [43]. Sha et al. [52] reported the effects of different temperature extraction (55, 65, and 75˚C) on the breaking position of the gelatin sequence segments, and the high temperature was an important factor in the breakdown into a smaller fragment.

Primary amino acid sequences of peptides affected gel strength in the collagen α-1 and α-2 chains. Type I collagen is the most abundant protein in animal hides. It forms a stable triple

helix consisting of three polypeptide chains: two identical strands and a genetically distinct strand. Each strand comprises more than 1000 amino acid residues with a repeating Gly-Xaa-Yaa unit, where Gly is glycine, and Xaa or Yaa can be any amino acid residue except trypto-phan. Differences in collagen type affect gel strength. The findings can be used to investigate unique sequences and detect multiple different types of protein from bovine, fish, and jellyfish gelatin, providing unambiguous identification of gel properties. The main difficulties encoun-tered were finding reliable gel strength of gelatin from the LC-MS analysis.

## Conclusion

The salt of salted jellyfish by-products was completely removed when the integration of con-ventional wash and ultrasonic wash operated at a fixed frequency (40 kHz) and power (220 W) for 100 min was applied. However, the duration of ultrasonication for 100 min caused the low-est water absorption and generated shrunk and broken pieces. For jellyfish gelatin production, the 40 min-ultrasonicated by-products subjected to hydrochloric acid pre-treatment and extracted at 80˚C for 4 h showed the best gel strength of 447 g. All jellyfish gelatins had a vis-cosity of 7cP and a melting point and gelling point of 15˚C and 8˚C, which is lower than BG and FG. The peptides determined by the LC-MS method display unique proteins and sequences of BG, FG, and JFG. The proteomic analysis showed that 21 peptides were similarly presented in JFG40-4, JFG100-8, FG, and BG. Proteins in jellyfish gelatins were more closely related to FG than BG. Of the 27 proteins in FG, the JFG40-4 also found 19 types of proteins. As a result, differences in proteins, sequences, and quantity of BG, FG, and JFG lead to differ-ent gelatin qualities. However, the analysis of the mechanism of jellyfish gelatin gel formation and its stability explained by complex peptides from the LC-MS method is still needed. Based on the results of this study, using ultrasound to partially eliminate salt residue from jellyfish by-products may be a method of choice in the preparatory step, and the extraction at 80˚C for 4 h for producing the highest gel strength of jellyfish gelatin.

## Supporting information

**S1 File.**
(DOCX)

## Acknowledgments

The authors would like to thank Dr. Philipus Pangloli from the Department of Food Science, University of Tennessee, for his support in laboratory analysis.

## Author Contributions

**Conceptualization:** Pokkwan Hutangura, Pisit Wongsa-Ngasri, Sittiruk Roytrakul, Benjawan Thumthanaruk.

**Data curation:** Artima Lueyot.

**Formal analysis:** Sittiruk Roytrakul, Benjawan Thumthanaruk.

**Funding acquisition:** Benjawan Thumthanaruk.

**Investigation:** Artima Lueyot, Vilai Rungsardthong, Pokkwan Hutangura, Sittiruk Roytrakul, Tao Wu, Benjawan Thumthanaruk.

**Methodology:** Artima Lueyot, Benjamaporn Wonganu, Savitri Vatanyoopaisarn, Pokkwan Hutangura, Pisit Wongsa-Ngasri, Sittiruk Roytrakul, Sawanya Charoenlappanit, Benjawan Thumthanaruk.

**Project administration:** Benjawan Thumthanaruk.

**Resources:** Benjawan Thumthanaruk.

**Software:** Sittiruk Roytrakul.

**Supervision:** Benjawan Thumthanaruk.

**Writing – original draft:** Artima Lueyot, Benjawan Thumthanaruk.

**Writing – review & editing:** Tao Wu, Benjawan Thumthanaruk.

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
