## [Decision Letter · Decision Letter 0]

25 Apr 2022

PONE-D-22-06129Improved jellyfish gelatin quality through ultrasound-assisted salt removal and an extraction processPLOS ONE

Dear Dr. Thumthanaruk,

Thank you for submitting your manuscript to PLOS ONE. After careful consideration, we feel that it has merit but does not fully meet PLOS ONE’s publication criteria as it currently stands. Therefore, we invite you to submit a revised version of the manuscript that addresses the points raised during the review process.

We look forward to receiving your revised manuscript.

Kind regards,

Roswanira Abdul Wahab

Academic Editor

PLOS ONE

Journal Requirements:

(Artima received funding from the Department of Agro-Industrial, Food and Environmental Technology and College of graduate, King Mongkut’s University of Technology North Bangkok. This work was financially supported by Thailand Science Research and Innovation (TSRI) (No. PHD59I0041))

Additional Editor Comments:

Dear author,

The reviewers have submitted their comments and concurred that the article requires a major revision.

Reviewers' comments:

Reviewer's Responses to Questions

**Comments to the Author**

1. Is the manuscript technically sound, and do the data support the conclusions?

Reviewer #1: Yes

Reviewer #2: Yes

2. Has the statistical analysis been performed appropriately and rigorously? 

Reviewer #1: Yes

Reviewer #2: No

3. Have the authors made all data underlying the findings in their manuscript fully available?

Reviewer #1: Yes

Reviewer #2: Yes

4. Is the manuscript presented in an intelligible fashion and written in standard English?

Reviewer #1: Yes

Reviewer #2: Yes

5. Review Comments to the Author

Reviewer #1: Review PONE-D-22-06129

The authors reported the ultrasound-assistant salt enhanced the removal and extraction of fish gelatin. The work is meaningful and helpful for corresponding food industry. However, the literature needs to be updated and the discussion should be enhanced.

Corresponding previous reports were missing. For instance, the ultrasound effects on fish gelatin extraction, 10.1111/jtxs.12408.

FTIR results: it was reported that salt and sugar caused peak wavenumber shift of fish gelatin (10.1016/j.foodhyd.2014.10.021). Thus, the corresponding discussion here needs to be improved. Whether the current salt extraction caused peak wavenumber shift? If not, why no influence?

The effects of ultrasound treatment should be discussed further. Through what effects did the ultrasound contribute the ‘assisted effects’?

The integration of the results from different parameters should be enhanced. They should be related to each other, which needs further elaboration.

Abstract: the authors may need to check with the journal requirement. In general, it might be too long for being a scientific abstract.

Lines 131-137: most of the description focused more on approaches but not objectives. The objectives should not be specific conditions like 5 different levels etc.

Unit: SI format should be applied. For instance, Line 152: change ‘minutes’ to ‘min’.

More references on fish gelatin’s structure and corresponding structural modification with other ingredients should be applied. The results and discussion were too much descriptive but lacked of in-depth work on the science behind these changes. The reference section needs to be updated and should focus more on mechanistic reports.

What’s the relationship between ultrasound effect and the salt effect? Synergistic, enhanced? Some previous reports indicated they might influence each other, for instance, 10.1039/c5ay00150a. How about the current effect?

Reviewer #2: Comments:

This manuscript was to investigate the ultrasonic-assisted salt removal from the salted jellyfish by-products to obtain the quality of gelatin extract. The abstract should be rewritten to be concise. Moreover, the methodology and discussion are needed to improve. Some comments are below.

Abstract

The abstract is too long. It should be concise and show only the importance of finding. Please to make it clear as well. Moreover, the duration of ultrasound treatment at 8 h (L. 48-50) is not correct. Please check.

Introduction

L. 103-108: Was the salt content in the salted jellyfish by-products 20-25%? Please clarify. Also, how long does it take for traditional desalination method? What was the conditions of traditional method?

Materials and methods

L. 141-148: Any change of tap water during overnight? Was the conventional washing process dried at the end? Please clarify.

L. 149-155: Why the authors said “Clean desalted jellyfish samples were….”? What did it mean? Please correct. What was the capacity of sonication bath and the size of sample used in each batch? What was the moisture content after drying?

L. 157-175: Why selected at 80oC with different extraction times to extract gelatin?

L. 188: Please check the Eq. (1). It is not a complete equation.

L. 197: Was it not covered 100 min? Please correct.

L. 199-201: Was the moisture content based on wet or dry bases? Please clarify.

L. 203-204: Please describe briefly.

L. 214: Why selected at 6.67% gelatin? Please clarify.

L. 215-217: Why used different magnification for raw materials and gelatin gels?

L. 224: What was ambient temperature?

L. 227-229: The weight of sample on L. 198 is indicated as Wo but why here used 200 in the equation, not as it was exactly weighted. Please explain.

L. 232: Please give the full name of GMIA.

L. 235-236: Please provide the method briefly.

L. 273-277: What was the experimental design used? How many treatments studied?

Results and discussion

L. 283: What were the raw salt contents of each treatment? Please clarify.

L 292-299: How possible to maintain the temperature in the sonicated bath? What was the temperature at 100 min of sonication time? Did the authors measure? Please explain.

Table 1: What were the original salt content of each treatment? Why the weight and moisture content of control showed 0.00? Why those were negative values at U80 and U100? Please explain.

L. 338-342-: Was the X-ray fluorescence studied in this study?

L. 349: The use of ultrasound helped reduce the washing time for how much? Please clarify.

L. 353-356: From Table S2, the gel strength of the control at JFG0-8H showed higher value which might not be the same of this explanation. Please explain.

L. 359-362: The wave numbers showed more than the number of treatments. Please clarify.

Table S2: Any significant differences among treatments of all data? Please add. Moreover, it should indicate in the explanation as well.

L. 432-434: What was the general gel strength from jellyfish by-products?

L 445-454, 467-482: Please report the results of each treatment.

L. 484-496: Please write the method in the materials and method section.

Conclusion

L. 536-539: Please revise. It is redundant.

6. PLOS authors have the option to publish the peer review history of their article (what does this mean?). If published, this will include your full peer review and any attached files.

Reviewer #1: **Yes: **Hongshun Yang

Reviewer #2: No

---

## [Author Response · Author response to Decision Letter 0]

23 Jul 2022

Dear Academic Editor and Reviewers, 

Below are the responses to the academic editor and reviewers' comments. All answers are in red. 

Journal Requirements:

Reviewer #1: Review PONE-D-22-06129

The authors reported the ultrasound-assistant salt enhanced the removal and extraction of fish gelatin. The work is meaningful and helpful for corresponding food industry. However, the literature needs to be updated and the discussion should be enhanced.

Corresponding previous reports were missing. For instance, the ultrasound effects on fish gelatin extraction, 10.1111/jtxs.12408.

FTIR results: it was reported that salt and sugar caused peak wavenumber shift of fish gelatin (10.1016/j.foodhyd.2014.10.021). Thus, the corresponding discussion here needs to be improved. Whether the current salt extraction caused peak wavenumber shift? If not, why no influence?

Ans. The FTIR results of the desalted jellyfish used ultrasonic-assisted washing shows a lower wavenumber at the amide III peak but the peak change because the ultrasonic wave disrupts the helical structure unstable. The new information has been added in L.412-421 "The amide III bands of the exposed samples to ultrasonic waves from 0, 20, 40, 60, 80, and 80 min appeared at 1244.59, 1243.77, 1241.31, 1238.85, 1236.38, and 1228.18 cm 1, respectively. Although, the wavenumber of characteristic spectral bands collagen (amide A, B, I, II, III) was shifted to lower frequencies in 100 mins ultrasonically treated sample. The longer ultrasonication duration might increase the thermal and unstable crosslink of protein molecules. However, a slight difference in the wavenumber of amide II and III can indicate that collagen structure might remain the same as in the untreated. Apart from the effect of ultrasonic waves on jellyfish collagen, the desalination effect of salted jellyfish by ultrasound also lowered the wavenumbers of the amide III band. Nadzrin et al. [45] reported the effect of salt content on the amide III band. The H+ of salt content can facilitate complexation with atoms in the amide III functional group, increasing ionic conductivities and lower wavenumber. 

The effects of ultrasound treatment should be discussed further. Through what effects did the ultrasound contribute the 'assisted effects'?

Ans. The ultrasound affects protein structure, and the discussion has been added in L.361-366 "The microstructure results of jellyfish by-products revealed the scar remaining on the jellyfish sample surface caused by the direct crystal salts added in the salting process. The effect of ultrasound frequency generates vibration waves that travel from the water to the protein surface and cause the detachment of salts from the surface of the jellyfish protein. The longer the ultrasound time, the greater the removal of salts from the protein (Table S2)." As a result, the ultrasound assists the salt solubility from jellyfish by-products. 

The integration of the results from different parameters should be enhanced. They should be related to each other, which needs further elaboration.

Ans. The feasible application of ultrasonic in jellyfish can cause structural changes to muscle tissue, thereby improving the salt removal and enhancing chemical diffusion behavior for gelatin extraction. The discussion has been added in L. 444-453. "The ultrasonic wave treatment primarily caused damage to the muscular structure and induced disruption of the helical structure of ultrasonicated jellyfish by-products [46]. When subjected to the high temperature, the molecular change from an α-helical structure to a random coil, showing collagen denaturation to gelatin [50]. Moreover, the reduction in the amides I and III band intensity by extraction time was associated with the more significant loss of molecular order in jellyfish gelatin. The collapsing cavitation bubbles lead to increased surface area and solvent absorption in the jellyfish tissue layer, enhancing the accessibility of the water as solvent resulting in increased extraction yield, which, in turn, lowers the gel strength. 

Abstract: the authors may need to check with the journal requirement. In general, it might be too long for being a scientific abstract.

Ans. Plos One allows the word count of abstract to 500 words. However, the revised abstract has been made with 347 words compared to the previous one of 489 words. 

Lines 131-137: most of the description focused more on approaches but not objectives. The objectives should not be specific conditions like 5 different levels etc.

Ans. L125-130. The previous statement was, "Therefore, the study aimed to eliminate the salt remaining in the cleaned desalted jellyfish by-products by varying 5 different duration times of ultrasound for 20, 40, 60, 80, and 100 min at a fixed frequency and power (40 kHz and 220 W) of a sonication bath. In addition, the duration time of extraction varied from 4, 6, and 8 h at 80oC. The change has been made to "Therefore, the objectives of this study were to determine the combined effect of traditional washing subsequently to ultrasonic treatment on the reduction of the salt content of desalted jellyfish by-products and for gelatin production, to determine the effect of extraction time of combined wash-ultrasonicated desalted jellyfish on gelatin gel qualities." 

Unit: SI format should be applied. For instance, Line 152: change 'minutes' to 'min'.

Ans. Line 148: 'minutes' has been changed to "min."

More references on fish gelatin's structure and corresponding structural modification with other ingredients should be applied. The results and discussion were too much descriptive but lacked of in-depth work on the science behind these changes. The reference section needs to be updated and should focus more on mechanistic reports.

Ans. The results and discussion have added more information in in-depth detail and are updated in the new reference section in structural modification in the FTIR part (L.403-459)

"The change of collagen secondary structure and hydrogen bonding between N-H stretch and C=O represent the amide I, II, and III at around 1660, 1550, and 1220 cm-1, respectively. The shift of the amide I band's wavenumbers dictates the triple helix's significant loss via the breaking down of H-bonds between α-chains [41], thereby changing the secondary structure of jellyfish collagen. Amide, I band from FTIR spectra, were slightly shifted, as shown in Fig 3A. It indicates that ultrasonication does not affect the crosslink between protein molecules. The change of amide II also dictates the secondary structure changes of collagen and the hydration of protein [43]. The amide II vibration mode is attributed to the peptide group's out-of-phase C-N stretch and in-plane N-H deformation modes. Bands of amide II were slightly decreased and observed at 1550.75, 1547.46, 1545.00, 1542.54, 1539.26, and 1536.79 cm-1 for treated with ultrasonic waves from 0, 20, 40, 60, 80, and 100 min, respectively. The amide III represents the combination peaks between C-N stretching vibration and N-H deformation from amide linkages and absorptions arising from wagging vibrations from CH2 groups of the glycine backbone proline sidechains, which are involved in the intermolecular interactions of collagen [44]. The amide III bands of the exposed samples to ultrasonic waves from 0, 20, 40, 60, 80, and 80 min appeared at 1244.59, 1243.77, 1241.31, 1238.85, 1236.38, and 1228.18 cm 1, respectively. Although, the wavenumber of characteristic spectral bands collagen (amide A, B, I, II, III) was shifted to lower frequencies in 100 mins ultrasonically treated sample. The longer ultrasonication duration might increase the thermal and unstable crosslink of protein molecules. However, a slight difference in the wavenumber of amide II and III can indicate that collagen structure might remain the same as in the untreated. Apart from the effect of ultrasonic waves on jellyfish collagen, the desalination effect of salted jellyfish by ultrasound also lowered the wavenumbers of the amide III band. Nadzrin et al. [45] reported the effect of salt content on the amide III band. The H+ of salt content can facilitate complexation with atoms in the amide III functional group, increasing ionic conductivities and lower wavenumber. 

The gelatin was extracted from ultrasonicated jellyfish by-products at 80oC for 4 h. The FTIR spectra of gelatin are shown in Fig 3B and Table S2. All jellyfish gelatins exhibit major absorption bands in the amide band region. The spectrum of amide I peak that appears at 1600-1700 cm-1 is mainly related to secondary [46,47]. The amide I band of JFG0-4 to JFG100-4 was shown at 1649.75, 1643.29, 1643.06, 1636.47, 1631.93, and 1629.65 cm-1, respectively. The high-temperature extraction might cause losses of molecular order of triple helix of collagen due to thermal uncoupling of inter-molecular crosslink. The collagen denaturation was interpreted as in disintegration of the triple helix collagen structure into a random coil [44].

The amide II bands were found at a spectrum between 1550-1600 cm-1. All the samples had a band with insignificant differences in the spectra of the amide II band. They were found at the wavenumbers of 1550.75, 1547.46, 1545.00, 1542.54, 1539.26, and 1536.79 cm-1, respectively. Amide II bands arise from the in-plane bending mode of NH of the peptide groups and stretching vibration of CN groups [46]. The lower wavenumber of JFG100-4 indicated more intensity of NH involved in hydrogen bonding with adjoining alpha chains [47]. The amide III region occurs in the range of 1220-1330 cm-1. The band at 1245-1270 cm-1 is attributed to random coil structure [48]. The vibration peak at near 1270 cm-1 is revealed the amide III bands of collagen, which is related to the triple helical structure of collagen [49]. The change in a band indicates the unstable helical structure, which affects weak gel properties. Prior to extraction, the ultrasonic wave treatment primarily caused damage of the muscular structure and induced disruption of the helical structure of ultrasonicated jellyfish by-products [46]. When subjected to the high temperature, the molecular change from an α-helical structure to a random coil, showing collagen denaturation to gelatin [50]. Moreover, the reduction in the amides I and III band intensity by extraction time was associated with the more significant loss of molecular order in jellyfish gelatin. The collapsing cavitation bubbles lead to increased surface area and solvent absorption in the jellyfish tissue layer, enhancing the accessibility of the water as solvent resulting in increased extraction yield, which, in turn, lowers the gel strength. 

The spectral region between 3400 - 3440 cm-1 corresponds to the band of amide A, whereas amide B corresponding to the asymmetric stretching vibration of = C-H and -NH3+ [5] was observed at 2931 – 2921.85 cm-1. The band shift of amide A and amide B to lower wavenumbers suggests a hydrogen bond's sensitivity with high-temperature extraction and the effect of high-temperature facilitating the interactions of -NH3+ groups between peptide chains."

What's the relationship between ultrasound effect and the salt effect? Synergistic, enhanced? Some previous reports indicated they might influence each other, for instance, 10.1039/c5ay00150a. How about the current effect?

Ans. The ultrasound effect has a synergistic and enhanced effect on desalination and jellyfish gelatin extraction. The collapsing cavitation bubbles lead to increased surface area and solvent absorption in the jellyfish tissue layer, enhancing the accessibility of the water as solvent resulting in increased extraction yield. The current effect, the ultrasound effect can remove salt content 100% on salted jellyfish, also had a high gel strength of jellyfish gelatin compared with control (untreated with ultrasound).

Reviewer #2: Comments:

This manuscript was to investigate the ultrasonic-assisted salt removal from the salted jellyfish by-products to obtain the quality of gelatin extract. The abstract should be rewritten to be concise. Moreover, the methodology and discussion are needed to improve. Some comments are below.

Ans. The abstract had been rewritten.

Abstract

The abstract is too long. It should be concise and show only the importance of finding. Please to make it clear as well. 

Ans. The abstract was revised, as seen below. 

Using by-products of salted jellyfish for gelatin production offers valuable gelatin products rather than animal feed. Several washes or washing machines have reported removing salt in salted jellyfish. However, the green ultrasound technique has never been reported for the desalination of salted jellyfish. The objectives were to determine how effectively the salted jellyfish raw materials salt removal was done by combining the traditional wash and then subjected to the ultrasonic waves in a sonication bath for 20-100 min. For gelatin production, the desalted jellyfish by-products were alkaline pretreated and extracted with hot water for 4, 6, and 8 h. Results showed that increased duration of ultrasound time increased the desalination rate. The highest desalination rate of 100% was achieved using 80 and 100 min ultrasonic time operated at a fixed frequency (40 kHz) and power (220 W). The jellyfish gelatin extracted for 4, 6, and 8 h showed gel strengths in 121-447, 120-278, and 91-248 g. The 80 min ultrasonicated sample and hot water extraction for 4 h (JFG80-4) showed the highest gel yield of 32.69%, with a gel strength of 114.92 g, but the 40 min ultrasonicated sample with 4 h of extraction delivered the highest gel strength of 447 g (JFG40-4) and the lower yield of 10.60%. The melting and gelling temperatures of jellyfish gelatin from ultrasonicated samples ranged from 15-25oC and 5-12oC, which are lower than bovine gelatin (BG) and fish gelatin (FG). Monitored by FITR, the synergistic effect of extended sonication time (from 20-100 min) with 4 h extraction time at 80oC caused Amide I, II, and III changes. Based on the proteomic results, the peptide similarity of JFG40-4 was 17, 23, or 20 peptides compared to either BG, FG, or JFG100-4. The 14 peptides were similarly found in all JFG40-4, BG, and FG. In conclusion, for the first time in this report, the improved jellyfish gel can be achieved when combined with traditional wash and 40 min ultrasonication of desalted jellyfish and extraction time of 4 h at 80oC.

Moreover, the duration of ultrasound treatment at 8 h (L. 48-50) is not correct. Please check.

Ans Line 47 "8 h" has been changed to "80 min"

Introduction

L. 103-108: Was the salt content in the salted jellyfish by-products 20-25%? Please clarify. Also, how long does it take for traditional desalination method? What was the conditions of traditional method?

Ans. No, the salt content of salted jellyfish is 16.0% in this study. The traditional desalination method takes around 22 h, including washing with water step for 3-4 h, then soaking in water overnight (12h) after that, using a heavy basket to push out of water for 4-6 h. The condition of the traditional method is salt content below 0%, determined by a salinometer.

Materials and methods

L. 141-148: Any change of tap water during overnight? Was the conventional washing process dried at the end? Please clarify.

Ans. The salted jellyfish was performed several washes and then overnighted soaking without changing the water for traditional washing. After desalting, the sample was dried at 60 oC for 24 h to reach moisture of 9.94% (Lueyot et al., 2021).

L. 149-155: Why the authors said "Clean desalted jellyfish samples were…."? What did it mean? Please correct. What was the capacity of sonication bath and the size of sample used in each batch? What was the moisture content after drying?

Ans. The phrase "Clean desalted jellyfish samples " means the raw material sample that was washed and then ultrasonicated. In this case, the phrase has been changed to ultrasonicated desalted jellyfish samples were wash jellyfish. The moisture content after drying is 9.94 %.

L. 157-175: Why selected at 80oC with different extraction times to extract gelatin?

Ans. Our previous optimization study found that the extraction condition of 81.4 oC for 8.1 h yielded 27.3% and gelatin gel strength of 284.14 g. Then, in this study, we simplified the extraction condition at 80 oC and varied the duration time to 4,6 and 8 h. (doi.org/10.21203/rs.3.rs-62849/v1). 

L. 188: Please check the Eq. (1). It is not a complete equation.

Ans. Change in salt content = (S0-Si)/S0*100 change into salt reduction (%).

Eq.(1) is salt content (%) = 5.8 x [(V1 x N1) – (V2 x N2)]/ W 

Eq.(2) is Salt reduction (%) = (S0-Si)/S0*100 

L. 197: Was it not covered 100 min? Please correct.

Ans. The 100 min has been written in Line 197.

L. 199-201: Was the moisture content based on wet or dry bases? Please clarify.

Ans. The moisture content was calculated based on a dry basis. 

L. 203-204: Please describe briefly.

Ans. The microstructure analysis of dried desalted jellyfish raw material was a preparation sample has been added in L.212-213 "For the surface morphology of dried desalted jellyfish sample each ultrasound treatment samples were coated using a gold-palladium alloy coater (JFC-1200 fine coater, JEOL, Japan).

L. 214: Why selected at 6.67% gelatin? Please clarify.

Ans. The gel strength determination was prepared at 6.67%, according to the standard GMIA method. 

L. 215-217: Why used different magnification for raw materials and gelatin gels?

Ans. The image of gelatin gel at 1,000 and 5,000 magnifications cannot see the structure and the image of jellyfish gelatin gel can observe at high magnification (10,000 and 30,000 magnification)

L. 224: What was ambient temperature?

Ans. The ambient temperature was 25±3 oC 

L. 227-229: The weight of sample on L. 198 is indicated as Wo but why here used 200 in the equation, not as it was exactly weighted. Please explain.

Ans. The weight of the sample on L.203 was used for calculated water absorption, and Wo = 200 g. The experiment used 200 g of desalted jellyfish from traditional washing and then put it in the ultrasound bath to remove salted content in jellyfish again. And the weight for calculated gelatin yield is 20g.

L. 232: Please give the full name of GMIA.

Ans. GMIA stands for the Gelatin Manufacturers Institute of America. 

L. 235-236: Please provide the method briefly.

Ans. The method has been added in Line 257-259. "The experiment was divided into two sections. The first section was sample digestion with trypsin, and the second was protein peptides identification by LC-MS/MS.

L. 273-277: What was the experimental design used? How many treatments studied?

Ans. The experimental ultrasound-assisted washing of jellyfish has one factor, and they have 6 treatments. The experimental design of jellyfish gelatin extraction is a factorial experiment design, and they have 2 parameters, including the time of ultrasonic material (6 treatments) and extraction time (3 treatments). The total extraction treatment is 18 treatments.

Results and discussion

L. 283: What were the raw salt contents of each treatment? Please clarify.

Ans. In this experiment, the sampling of desalted jellyfish by-products was random and showed a salt content of 16.0 %.

L 292-299: How possible to maintain the temperature in the sonicated bath? What was the temperature at 100 min of sonication time? Did the authors measure? Please explain.

Ans.: This experiment used a sample of 200 g the experiment. The experiment does not maintain the temperature. The temperature of the water medium increases by 1oC every 5 min of exposure time. The temperature at the beginning is 28 oC and the temperature at 100 min of sonication time is 54 oC.

Table 1: What were the original salt content of each treatment? Why the weight and moisture content of control showed 0.00? Why those were negative values at U80 and U100? Please explain.

Ans. The original salt content of salted jellyfish is 16 % (measurement by titration method) after washing with water several times until salt content is below 0 %, determined by the salinometer (control). The original salt content of each treatment is 1.16% before ultrasound treatment. The moisture content has negative values at U80 and U100 because the weight after ultrasonic treatment decreases when the treatment has a long time.

L. 338-342-: Was the X-ray fluorescence studied in this study?

Ans. L327: No. This study did not perform X-ray fluorescence. The results of salt were obtained from Charoenchokpanich et al. (30).

L. 349: The use of ultrasound helped reduce the washing time for how much? Please clarify.

Ans. L 375-380: The sentence stated, "Therefore, using ultrasound to eliminate salts in the washing benefits to shorten the washing time and remove salts in the sample". Several washes and overnight soaking did the tradition of salt removal. Instead of overnight soaking (approx. 15 h), ultrasonic was used in this study. The time was short for at least 13 h.

L. 353-356: From Table S2, the gel strength of the control at JFG0-8H showed higher value which might not be the same of this explanation. Please explain.

Ans. Our previous optimization study, the JFG0-8, discovered that extraction conditions of 81.4 °C for 8.1 h yielded 27.3% and 284.14 g of gelatin gel strength. Then, in this study, the extraction time was reduced to 4 and 6 hours. Reducing the extraction time for jellyfish gelatin is expected to improve the gel properties.

L. 359-362: The wave numbers showed more than the number of treatments. Please clarify.

Ans. The wave numbers in Line 359-362 have 6 treatments, and the result shows the wave number of all 6 treatments.

Table S2: Any significant differences among treatments of all data? Please add. Moreover, it should indicate in the explanation as well.

Ans. Table S2 has added the significant differences among treatments of all data and the Table S2 changes in Table S3.

L. 432-434: What was the general gel strength from jellyfish by-products?

Ans. The gel strength of jellyfish had between 108-118 g (Chancharern et al., 2016 and Rodsuwan et al., 2016)

L 445-454, 467-482: Please report the results of each treatment.

Ans: L. 513-521 has been changed to the value of each treatment. The gel strengths of gelatin extracted for 4 h of JFG0-4, JFG20-4, JFG40-4, JFG60-4, JFG80-4, and JFG100-4 were 129.62±1.10, 160.05±1.63, 447.01±1.06, 283.29±2.36, 278.71±2.90 and 121.66±2.27 g, respectively. The JFG0-6, JFG20-6, JFG40-6, JFG60-6, JFG80-6 and JFG100-6 for extraction time 6 h were 188.70±1.79, 246.76±1.98, 278.91±1.59, 191.98±3.04, 150.33±2.95 and 120.06±1.10 g, respectively. And extraction time for 8 h of JFG0-8, JFG20-8, JFG40-8, JFG60-8, JFG80-8 and JFG100-8 were 248.12±3.60, 207.75±2.55, 181.57±1.42, 152.95±2.88, 114.92±1.85 and 91.89±1.74, respectively.

L. 484-496: Please write the method in the materials and method section.

Ans. For identifying peptides for different bovine, fish, and jellyfish gelatin, we have the method in the materials and method section (Preparation of jellyfish gelatin peptide and identification by LC-MS/MS) added in L. 285-296. The MaxQuant (version 1.6.6.0) was used to quantify individual samples bioinformatically, and their MS/MS spectra were matched to the UniProt database using the Andromeda search engine [37]. The protein sequences were searched in FASTA files. The protein database was downloaded from the UniProt database of fish gelatin, bovine gelatin, bovine collagen, and alpha 2 and 1 chain. The MaxQuant ProteinGroups.txt file was loaded into Perseus version 1.6.6.0, and potential contaminants that did not correspond to any UPS1 protein were removed from the data set [38]. Max intensities were log2 transformed, and pairwise comparisons between conditions were made via t-tests. Perseus also imputed missing values using a constant value (zero). The visualization and statistical analyses were conducted using a MultiExperiment Viewer (MeV) in the TM4 suite software [10]. The Venn diagram displays the differences between protein lists originating from different samples.

Conclusion

L. 536-539: Please revise. It is redundant.

Ans. The previous statement was that 21 peptides were similarly presented in JFG40, FG, and BG. Proteins in jellyfish gelatins were more closely related to FG than BG. Of the 27 proteins in FG, 19 were also found in JFG 40. Line 635-638 has been revised to "The proteomic analysis shows that 21 peptides were similarly presented in JFG40, FG, and BG. Proteins in jellyfish gelatins were more closely related to FG than BG. Of the 27 proteins in FG, the JFG-40 also found 19 types of proteins."

If you have any concerns, please feel free to contact me.

Best regards, 

Benjawan Thumthanaruk 

Corresponding author

---

## [Decision Letter · Decision Letter 1]

17 Aug 2022

PONE-D-22-06129R1Improved jellyfish gelatin quality through ultrasound-assisted salt removal and an extraction processPLOS ONE

Dear Dr. Thumthanaruk,

Thank you for submitting your manuscript to PLOS ONE. After careful consideration, we feel that it has merit but does not fully meet PLOS ONE’s publication criteria as it currently stands. Therefore, we invite you to submit a revised version of the manuscript that addresses the points raised during the review process.

The reviewer has pointed out some issues on the revised article. Therefore, you are encouraged to address each issue point-by-point, in your revised article.

We look forward to receiving your revised manuscript.

Kind regards,

Roswanira Abdul Wahab

Academic Editor

PLOS ONE

Journal Requirements:

Reviewers' comments:

Reviewer's Responses to Questions

**Comments to the Author**

1. If the authors have adequately addressed your comments raised in a previous round of review and you feel that this manuscript is now acceptable for publication, you may indicate that here to bypass the “Comments to the Author” section, enter your conflict of interest statement in the “Confidential to Editor” section, and submit your "Accept" recommendation.

Reviewer #1: All comments have been addressed

Reviewer #2: All comments have been addressed

2. Is the manuscript technically sound, and do the data support the conclusions?

Reviewer #1: Yes

Reviewer #2: Yes

3. Has the statistical analysis been performed appropriately and rigorously? 

Reviewer #1: Yes

Reviewer #2: Yes

4. Have the authors made all data underlying the findings in their manuscript fully available?

Reviewer #1: Yes

Reviewer #2: Yes

5. Is the manuscript presented in an intelligible fashion and written in standard English?

Reviewer #1: Yes

Reviewer #2: Yes

6. Review Comments to the Author

Reviewer #1: Review PONE-D-22-06129R1

The authors have addressed the question quite well. The manuscript has been improved significantly. The current version is acceptable for publication.

Reviewer #2: Comments:

This manuscript is improved and the authors answer my questions properly. However, I have a quick question why the data on salt reduction in Table 1 of revised manuscript are different from the original manuscript, please explain. Some information should be added in the manuscript. For example, the ambient temperature should be indicated in the manuscript so the readers will understand the work better.

7. PLOS authors have the option to publish the peer review history of their article (what does this mean?). If published, this will include your full peer review and any attached files.

Reviewer #1: **Yes: **Hongshun Yang

Reviewer #2: No

---

## [Author Response · Author response to Decision Letter 1]

30 Aug 2022

Dear Academic Editor and Reviewers, 

Below are the responses to the academic editor and reviewers' comments. All answers are in red. 

Journal Requirements:

Ans. The reference has been checked to complete and correct.

Reviewer #2: Review PONE-D-22-06129

This manuscript is improved and the authors answer my questions properly. However, I have a quick question why the data on salt reduction in Table 1 of revised manuscript are different from the original manuscript, please explain. Some information should be added in the manuscript. For example, the ambient temperature should be indicated in the manuscript so the readers will understand the work better.

Ans : The data in Table 1 of revised manuscript has been recalculated using the equation in Line 175-179, including 

Eq.(1) is salt content (%) = 5.8 x (V1 x N1) – (V2 x N2/ W) 

Eq.(2) is salt reduction (%) = (S0-Si)/S0*100 

In this case, the salt content of each condition (U0, U20, U40, U60, U80, and U100) was initially measured. After calculation by using Eq. (2), salt reduction with respect to the control displayed the clear reducing trend of salt content. This is quite different from the previous data in original manuscript that reported the increased or decreased weight (g) of the sample after sonication. These raw data were hardly to explain the effect of sonication on salt content. Therefore, the additional sentence has been added in Line 298-300 that is “The salt content of each condition (U0, U20, U40, U60, U80 and U100) was initially measured ” (Table S1), and the salt reduction compared to the control (U0) was then calculated”. 

The temperature of the water medium was 25 ± 3oC (ambient temperature) has been added in Line 311.

If you have any concerns, please feel free to contact me.

Best regards, 

Benjawan Thumthanaruk 

Corresponding author

---

## [Decision Letter · Decision Letter 2]

28 Sep 2022

Improved jellyfish gelatin quality through ultrasound-assisted salt removal and an extraction process

PONE-D-22-06129R2

Dear Dr. Thumthanaruk,

We’re pleased to inform you that your manuscript has been judged scientifically suitable for publication and will be formally accepted for publication once it meets all outstanding technical requirements.

Kind regards,

Roswanira Abdul Wahab

Academic Editor

PLOS ONE

Additional Editor Comments (optional):

Reviewers' comments:

Reviewer's Responses to Questions

**Comments to the Author**

1. If the authors have adequately addressed your comments raised in a previous round of review and you feel that this manuscript is now acceptable for publication, you may indicate that here to bypass the “Comments to the Author” section, enter your conflict of interest statement in the “Confidential to Editor” section, and submit your "Accept" recommendation.

Reviewer #2: All comments have been addressed

2. Is the manuscript technically sound, and do the data support the conclusions?

Reviewer #2: Yes

3. Has the statistical analysis been performed appropriately and rigorously? 

Reviewer #2: Yes

4. Have the authors made all data underlying the findings in their manuscript fully available?

Reviewer #2: Yes

5. Is the manuscript presented in an intelligible fashion and written in standard English?

Reviewer #2: Yes

6. Review Comments to the Author

Reviewer #2: The authors clarify and explain my questions properly. Now it is suitable and accepted for publication.

7. PLOS authors have the option to publish the peer review history of their article (what does this mean?). If published, this will include your full peer review and any attached files.

Reviewer #2: No

---

## [Editor Report · Acceptance letter]

6 Oct 2022

PONE-D-22-06129R2 

Improved jellyfish gelatin quality through ultrasound-assisted salt removal and an extraction process 

Dear Dr. Thumthanaruk:

I'm pleased to inform you that your manuscript has been deemed suitable for publication in PLOS ONE. Congratulations! Your manuscript is now with our production department. 

Kind regards, 

on behalf of

Dr. Roswanira Abdul Wahab 

Academic Editor

PLOS ONE